# Topologically protected modes in non-equilibrium stochastic systems

Arvind Murugan[1,2] & Suriyanarayanan Vaikuntanathan[1,3]

Non-equilibrium driving of biophysical processes is believed to enable their robust functioning despite the presence of thermal fluctuations and other sources of disorder. Such robust functions include sensory adaptation, enhanced enzymatic specificity and maintenance of coherent oscillations. Elucidating the relation between energy consumption and organization remains an important and open question in non-equilibrium statistical mechanics. Here we report that steady states of systems with non-equilibrium fluxes can support topologically protected boundary modes that resemble similar modes in electronic and mechanical systems. Akin to their electronic and mechanical counterparts, topological-protected boundary steady states in non-equilibrium systems are robust and are largely insensitive to local perturbations. We argue that our work provides a framework for how biophysical systems can use non-equilibrium driving to achieve robust function.

[1] James Franck Institute, University of Chicago, Chicago, Illinois 60637, USA. [2] Department of Physics, University of Chicago, Chicago, Illinois 60637, USA. [3] Department of Chemistry, University of Chicago, Chicago, Illinois 60637, USA. Correspondence and requests for materials should be addressed to S.V. (email: svaikunt@uchicago.edu).

Understanding the tradeoffs between energy consumption and organization in far-from-equilibrium soft matter systems remains a challenging problem in statistical mechanics[1,2]. Mechanisms used by biological systems to process information and achieve ordered states are far from equilibrium and require energy dissipation[3]. Kinetic proofreading mechanisms used by the cell to ensure high fidelity copying of genetic material use futile energy consuming cycles to decrease the error rates in DNA replication[4]. Non-equilibrium forces have also been implicated in the functioning of biochemical networks responsible for adaptation[5], ultra-sensitivity[6,7] and timing of events in cell cycle[8].

While the behaviour and characteristics of equilibrium systems—where no energy is dissipated—are well known, general principles governing the steady state or fluctuations in it in far-from-equilibrium conditions are just being discovered[9]. Here we show that the steady states of non-equilibrium stochastic systems can be localized and have a character similar to topologically protected boundary modes in mechanical and electronic systems[10–13]. Similar to topological boundary modes in electronic and mechanical systems, we find that non-equilibrium systems that support topologically protected steady states have properties that are robust and generally insensitive to local perturbations. This surprising connection between classical non-equilibrium stochastic systems and topologically non-trivial mechanical and electronic systems constitutes our main result. Our results provide a framework for understanding how non-equilibrium forces can be tuned so that the steady state density is preferentially driven to and robustly localized in a desired region of phase space.

## Results

**Topological protection in Markov state networks**. We derive and illustrate our results using idealized non-equilibrium Markov state models of kinetic proofreading networks[14], sensory adaptation networks[5] and other biophysical processes. Topologically non-trivial electronic systems and meta-materials are composed of bulk regions formed by periodic replication of a certain unit cell. These bulk regions terminate at an 'interface'. Non-equilibrium biochemical reactions can be modelled as a Markov state process whose rate constants break detailed balance. Borrowing set-up and terminology from topologically non-trivial electronic systems, we will also imagine biochemical reaction networks that are composed of bulk-like regions. In these regions, mesoscopic chemical states represented in the Markov state process and the links denoting transitions between them are periodically replicated. Bulk-like regions terminate at interfaces where translational symmetry is lost. We show that when a biochemical network can be decomposed into two ordered bulks that meet at a possibly disordered interface, the ordered bulks can be each associated with a topologically invariant winding number. If the winding numbers are mismatched, we are guaranteed that the steady state distribution is localized at the interface between the bulks, even in the presence of strong disorder in reaction rates. In these biophysical examples, topologically protected localized steady states allow the system to robustly perform their desired information processing function[15].

To be precise, we derive our central results in the context of the Markov state model in Fig. 1a. The Markov state model is composed of two translationally invariant bulk-like regions with an interface connecting them. The rates of transitions in the bulk regions do not depend on the position along the horizontal axis (Fig. 1). The rates in the interfacial region interpolate between the two bulks. The spatial connectivity and structure of this Markov state network resembles that of networks routinely used to study

adaptation[5], kinetic proofreading[14] and cell-signal sensing[16]. As we argue later, these and other Markov state representations of biophysical processes can often be decomposed into bulk-like subgraphs stitched together by interfaces. The subgraphs themselves are formed by finite periodic replication of a particular module or motif. The construction of the idealized network in Fig. 1a was motivated by these features.

The dynamics of the system in Fig. 1a can be modelled using a master equation,

$$\frac{\partial \mathbf{p}}{\partial t} = W\mathbf{p} \qquad (1)$$

where the vector $\mathbf{p}$ contains the probability of occupancy of various nodes in the network and $W$ is a state-to-state transition matrix[17]. We are interested in conditions under which the steady state probability specified by equation (1) is localized at the interface between the two bulk regions. Surprisingly, we will find that localization of the steady state probability at the interface of the spatially heterogenous network in Fig. 1a can be predicted by assigning particular topological numbers to the bulk networks. The details of the interface are not relevant to the existence of these localized modes. The steady state behaviour is simply determined by the topological numbers.

To establish our central results, we find it convenient to consider the statistics of probability current along the horizontal axis in the network in Fig. 1. For that, we construct the closely related tilted current matrix $W(\lambda)$[18] with elements

$$W(\lambda)_{i,j} = W_{i,j} e^{\lambda(i_x - j_x)} \qquad (2)$$

where $i_x$ denotes the location of the node $i$ along the horizontal axis. The largest eigenvalue of $W(\lambda)$, $e(\lambda)$, is the cumulant generating function for currents along the horizontal axis, $J$ in the network[18,19]. In particular, $\frac{de(\lambda)}{d\lambda}\big|_{\lambda=0} = -\langle J \rangle$, giving the net average macroscopic current. To ensure the possibility of a non-zero current along the horizontal axis, we assume periodic boundary conditions and link up the left and right bulk networks through a second interface for our theoretical analysis.

Translational symmetry in the bulk regions makes it convenient to study their properties in terms of Fourier transforms. Specifically, we will imagine constructing translationally symmetric tilted current matrices, denoted by $W_{L/R}(\lambda)$, that describe the left (L) and right (R) bulk regions. A topological characterization of the bulk region can be obtained by first computing the determinant, $D(k, \lambda)$, of the Fourier transformed version of the translationally symmetric matrices $W_{L/R}(\lambda)$. Here $0 < k < 2\pi$ denotes the wave vector, and the determinant $D(k, \lambda) = |D(k, \lambda)|\exp(i\theta(k, \lambda))$ is a complex number with phase $\theta(k, \lambda)$. The determinant is periodic in $k$, $D(k + 2\pi, \lambda) = D(k, \lambda)$, by construction. A topological number can be assigned to the bulk network by determining the winding number $w$ of the phase $\theta(k, \lambda)$[12].

$$w = \frac{\theta(k + 2\pi, \lambda) - \theta(k, \lambda)}{2\pi} \qquad (3)$$

The winding number is one when $\theta(k, \lambda) = \theta(k + 2\pi, \lambda) + 2\pi$ and zero when $\theta(k, \lambda) = \theta(k + 2\pi, \lambda)$. We will show that the steady state probability distribution is localized when the bulk winding numbers are mismatched in an interval of $\lambda$ around zero, $\delta w \equiv w_L - w_R \neq 0$ for $\lambda$ in $\lambda^- < 0 < \lambda^+$ (Fig. 1). This remarkable topological constraint for the localization of the steady state probability at the interface constitutes our main theoretical result.

**A topological count for the number of localized eigenmodes**. To establish the localized nature of the steady state of the master equation (1), we will consider the eigenvectors of the tilted matrix

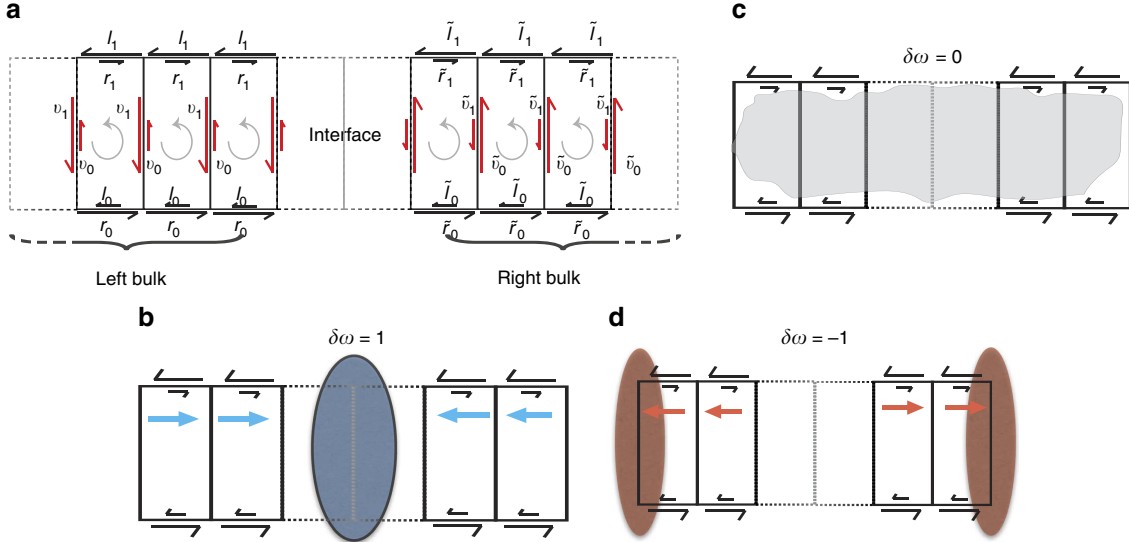

**Figure 1 | Topological protection in a model non-equilibrium Markov state network.** (**a**) A Markov state network with two translationally invariant periodic bulk regions connected together by an interfacial region. Analogous to edge modes in topological insulators and meta-materials, we show that the steady state probability distribution of these Markov state networks is determined by topological winding numbers, defined in equation (3), assigned to the bulk domains. (**b,d**) The steady state distribution is localized at the boundaries of the bulk domains when the winding numbers of the two bulk phases are mismatched, $\delta w \neq 0$. We call such states topologically protected states. (**c**) The steady state is not protected when $\delta w = 0$.

$W(\lambda)$. We will first show that the number of eigenvectors of $W(\lambda)$ with an eigenvalue of zero, henceforth referred to as zero modes, localized at the boundary between the left and right bulk networks is related to topological invariants computed in the bulk regions. We will then use this result to identify conditions under which $W(\lambda)$ has localized zero modes for values of $\lambda$ around $\lambda = 0$, $\lambda^- < 0 < \lambda^+$.

Such zero modes of $W(\lambda)$ imply that the steady state of the master equation supports a zero current along the horizontal axis $(J = 0)$[18]. The master equation transition matrix $W$ for the network in Fig. 1 is composed of two translationally invariant bulks. As we demonstrate in the Supplementary Note 4, the condition of zero current in the bulk regions of a topologically non-trivial network necessarily implies that the steady state probability of the master equation is exponentially localized.

To understand the topological nature of zero modes of $W(\lambda)$ at the boundary between distinct phases, we consider its local index[10],

$$\operatorname{ind} W(\lambda) \equiv \dim \ker \rho W(\lambda) - \dim \ker \rho W^{\mathrm{T}}(\lambda) \qquad (4)$$

where the matrix diagonal matrix $\rho$ has non-zero elements $\rho_{i,i} = 1$ for nodes $i$ in the interface boundary region[10] (Fig. 1), and dim $\ker \rho W(\lambda)$ denotes the dimensionality of the local kernel of $W(\lambda)$. dim $\ker \rho W(\lambda)$ is non-zero if the matrix $W(\lambda)$ has atleast one zero eigenmode contained in the interfacial region defined by $\rho$.

While the kernel of $W(\lambda)$ directly corresponds to zero modes of interest, the zero modes of $W^{\mathrm{T}}(\lambda)$ also have physical significance. Specifically, the elements of these zero modes, $f_i$, are equal to

$$f_i = \lim_{\tau \to \infty} \langle \exp(-\lambda J) \rangle_i \qquad (5)$$

where $\langle \dots \rangle_i$ is the average of trajectories, over a long time $\tau$, evolving according to equation (1) conditioned on them beginning at $i$ (ref. 18). In other contexts, it has been demonstrated that the eigenvectors of the adjoint of the master equation rate matrix $W^{\mathrm{T}}$ posses information related to the statistics of the first passage times and other such dynamical features[17].

As shown in the Supplementary Note 2, the index of $W(\lambda)$ can be expressed in terms of topological properties. Specifically, for a system with two bulk regions (as in Fig. 1), we find that ind $W$ is given by the difference of two numbers $w_{\mathrm{L}}$ and $w_{\mathrm{R}}$ computed in the left and right bulk phases of the network, respectively,

$$\operatorname{ind} W = \delta w \equiv w_{\mathrm{L}} - w_{\mathrm{R}} \qquad (6)$$

where,

$$w_{\mathrm{L/R}} \equiv \frac{1}{2\pi i} \int_0^{2\pi} \mathrm{dk} \partial_k \ln \left[ \det \left[ W_{\mathrm{L/R}}(\lambda, k) \right] \right] \qquad (7)$$

and $W_{\mathrm{L/R}}(\lambda, k)$ is the Fourier transform of the tilted translationally symmetric bulk transition matrix (Supplementary Note 2). The determinant $\det[W_{\mathrm{L/R}}(\lambda, k)]$ maps the Fourier transformed matrix to the complex plane, $\det[W_{\mathrm{L/R}}(\lambda, k)] = |\det[W_{\mathrm{L/R}}(\lambda, k)]| \exp(i\theta(\lambda, k))$. The numbers $w_{\mathrm{L/R}}$ in equation (7) hence simply compute the winding number of the phase $\theta(\lambda, k)$ (defined in equation (3)) as $k$ is varied from 0 to $2\pi$. In the SI, we present both an explicit proof for our central results using a network with a topology similar to that considered in Fig. 1 (Supplementary Note 2), and a proof based on ref. 10 (Supplementary Note 1).

The above equations predict the existence of boundary zero modes based on the spectrum of $W(\lambda)$ in the bulk alone; when $\delta w \neq 0$, $W(\lambda)$ must have a zero mode. In Supplementary Notes 3 and 4, we show that if, in fact, $\delta w \neq 0$ for an interval of $\lambda$ around $\lambda = 0$, $\lambda^- < 0 < \lambda^+$, then the highest eigenvalue of $W(\lambda)$, $e(\lambda)$ is constrained to be zero and the steady state probability distribution of the master equation is localized at the interface. Thus the edge modes of $W(\lambda)$ can be simply predicted by computing the winding numbers in the bulk of the networks.

To be concrete, we now numerically demonstrate these behaviours. In Fig. 2, we highlight the steady state probability distribution (ellipses) and the steady state distribution of its conjugate defined in equation (5) for parameters that ensure a winding number mismatch of $\delta w = 1$. In accordance with the theoretical predictions, the steady state probability distribution is localized at the interface between the two networks, while its conjugate defined by equation (5) is localized away from this interfacial region.

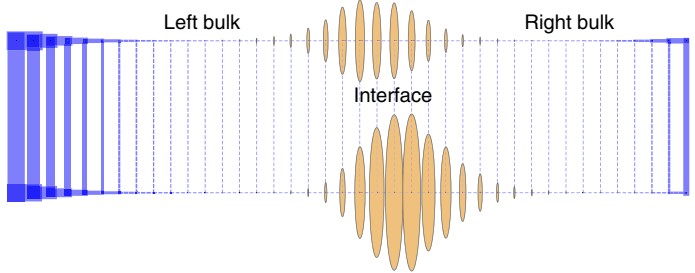

**Figure 2 | Numerical results from the ladder network for winding number mismatch within $\lambda^- < \lambda < \lambda^+$.** Node weights (orange) are proportional to the magnitude of the elements of the largest eigenvector of $W(\lambda)$ and vertical link weights (blue) are proportional to the corresponding elements of the largest eigenvector of $W^{\mathrm{T}}(\lambda)$.

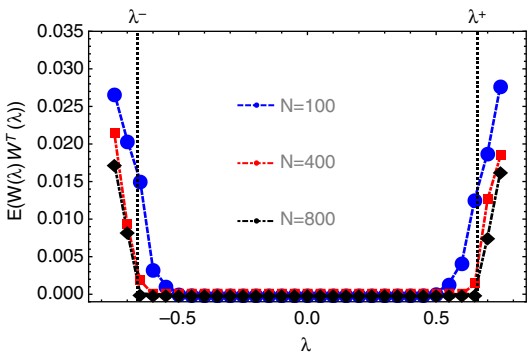

**Figure 3 | Lowest eigenvalue by magnitude of the operator $W(\lambda)W^{\mathrm{T}}(\lambda)$.** The dotted lines are the theoretical estimate for the region $\lambda^- \leq \lambda \leq \lambda^+$ within which $W(\lambda)W^{\mathrm{T}}(\lambda)$, and hence $W(\lambda)$, has a zero eigenmode. The agreement between numerical and theoretical results improves as a function of system size.

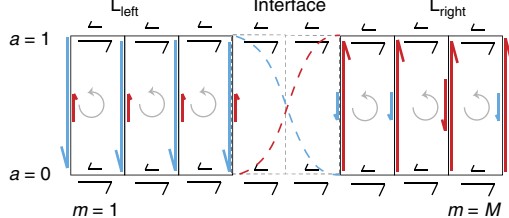

**Figure 4 | An idealized biophysical model of chemosensory adaptation.** The rates of transition between the active ($a = 1$) and inactive ($a = 0$) states are sigmoid functions of the methylation levels ($m$). The minimal biophysical model can be viewed a combination of two bulks with an interface between them.

In Fig. 3, we identify values of $\lambda$ for which $W(\lambda)$ has a zero eigenvalue for networks of various sizes. The numerical bounds $\lambda^-$ and $\lambda^+$ for which $e(\lambda)$ is constrained to zero agrees with the theoretical predictions obtained from the winding number analysis. The agreement is even more remarkable given that the numerical results were obtained from networks with quenched disordered. The topological connection allows us to predict the fluctuations of large complex non-equilibrium networks by performing a simple calculation in the bulk regions. The existence of the points $\lambda^+$ and $\lambda^-$ has further physical significance. Since the parameter $\lambda$ is coupled to the rates of transition along the horizontal axis, $\lambda \pm$ are related to the effective localization lengths for the probability distributions $\eta^{\mathrm{L/R}}$ (Supplementary Note 5),

$$\eta^{\mathrm{L/R}} \sim 1/\left(\lambda^\pm\right). \qquad (8)$$

The presence of topologically protected edge modes in other contexts is signalled by a gap between the zero energy state and the rest of the energy spectrum of the Hamiltonian operator[10,11]. We observe a similar connection in numerical simulations with finite-sized dynamical matrices $W(\lambda)$ (Supplementary Discussion 1). Since the non-zero eigenvalues of the master equation rate matrix set relaxation timescales, the band gap in the topologically non-trivial regime implies that perturbations away from the localized steady state are suppressed in a finite time $\tau$ related inversely to the band gap. This feature provides another basis for robustness.

**Localization and robustness in biophysical networks.** Chemosensory adaptation, kinetic proofreading and many other

information processing mechanisms in biology operate far from equilibrium[5,16]. Topologically protected modes in such networks can enable robust functioning in the presence disorder in the kinetic rates of the network. We first consider a commonly used idealized biophysical model[20] for chemotaxis adaptation in *Escherichia coli*. The dynamics of chemotaxis adaptation in *E. coli* can be described by specifying the activity $a$, and methylation level $m$ of the concentration sensing complex of proteins[15]. Transitions between the various mesoscopic states of the protein complex are governed by the Markov state model described in Fig. 4.

As discussed in the methods section, the transition rates along the activity axis generically are sigmoid functions of the methylation level $m$. The crossover region of these sigmoid functions is set by the chemoattractant concentration sensed by the protein complex. Further, in this class of idealized models, the rates of transition along the methylation axis are independent of methylation level and chemoattractant concentration[15]. This generic sigmoidal profile for the rates of transitions along the activity axis, and the methylation level independent kinetics along the methylation axis establishes the similarity between the chemotaxis adaptation network in Fig. 4 and the network we constructed in Fig. 1a. The minimal biophysical model for adaptation can hence be viewed as a combination of two periodic bulk networks with an interface between them. The location of the interface is set by the chemoattractant concentration.

In Fig. 5, we provide numerical results obtained from simulations with $N = 48$ methylation levels[21,22] (In Supplementary Fig. 6, we present numerical results from simulations with networks containing $10 \leq N \leq 48$. The theoretical predictions impose constraints on fluctuations in all these networks). Quenched disorder was introduced in the rates of the kinetic network. The parameter $S$ is a logarithmic function of the chemoattractant concentration. We find that the probability density is localized along the methylation axis, $p(m) \sim \exp(-|m - m_0|/\eta)$, where $p(m)$ denotes the probability

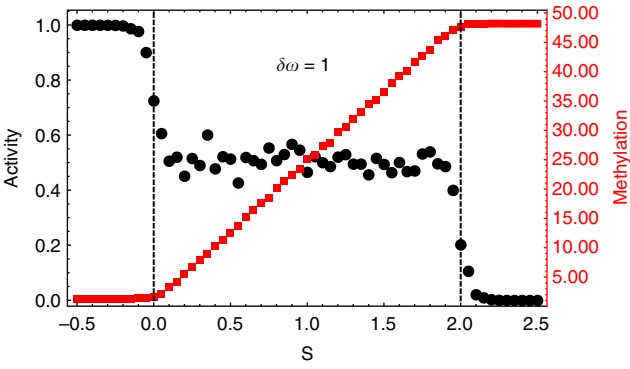

**Figure 5 | Steady state behaviour of the adaptive network.** The parameter $S$ is related to the logarithm of the chemoattractant concentration. The steady state probability density is localized along the horizontal methylation axis whenever $\delta w = 1$ and consequently the average methylation level (orange) tracks the chemoattractant concentration. In this regime, the activity of the network (black) is maintained at a set point over the same wide range of $S$.

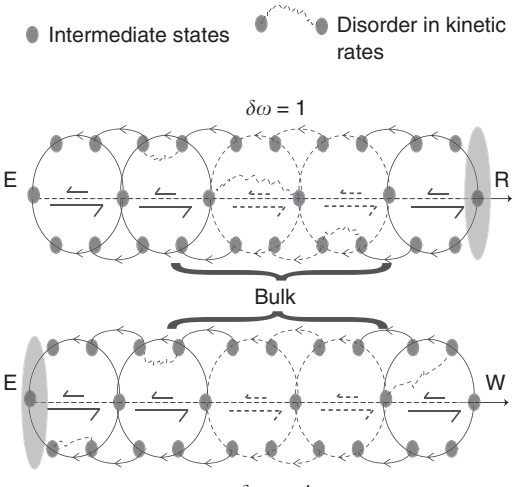

**Figure 6 | Markov state representation of kinetic proofreading mechanisms.** These networks can be viewed as a combination of a periodic bulk phase, which terminates at the right (R, top)/wrong (W, bottom) products on one end and the reactants (E) on the other end. Efficient and robust proofreading can be achieved by tuning the winding numbers of the bulk networks so that the probability distribution is localized at the products end for R and the reactants end for W.

of observing a methylation level $m$, $m_0$ is determined by the ligand concentration and $\eta$ is a localization length, whenever $\delta w = 1$. A similar form for $p(m)$ was derived in ref. 20 for a specific model of chemotaxis adaptation. Our framework generalizes these results. The average methylation level tracks the chemoattractant concentration in the regime where the winding number mismatch $\delta w = 1$ is maintained.

A robust adaptive network is defined as one that maintains a set activity of $a = a_0$ for a wide range of chemoattractant concentrations (and other perturbations). In Supplementary Fig 6, we show that chemoattractant-independent activity is ensured as long as the topological constraint $\delta w = 1$ is satisfied and the probability density is localized along the methylation axis as specified above. Further, the response of the system is insensitive to disorder in the kinetic rates in this topologically protected regime.

When the winding number mismatch is $\delta w = -1$, fluctuations in methylation levels are governed by the distribution, $p(m) \sim k_1 \exp(m/\eta_1) + k_2 \exp((M-m)/\eta_2)$, where $k_1$ and $k_2$ are functions of the parameters of the network and $\eta_{1,2}$ are localization lengths. The mean activity is not ligand independent (Fig. 5) in this regime. A topological transition separates the regime exhibiting robust adaptation from the regime in which adaptation is not achieved.

Topologically protected localized modes can promote robustness of kinetic proofreading, a non-equilibrium mechanism that enhances enzymatic specificity[4]. An enzyme $E$ might be faced with a substrate $R$ that is meant to be processed into a product but is hindered by the presence of a chemically similar undesirable substrate $W$. Previous literature has investigated many differing models and assumptions on the kinetics of $R$ and $W$ and driving forces that lead to differing localization and hence proofreading[3,14,23,24]. An intuitive way to understand proofreading is through localization (Fig. 6); despite substrates $R$ and $W$ having very similar kinetics when binding with an enzyme $E$, reactions with desired substrate $R$ should be localized near products, while reactions with $W$ should be localized near reactants[14,25]. In contrast, our results on topological protection provide a simple necessary and sufficient condition for efficient proofreading. We view the proofreading network as one bulk phase with one set of kinetics when the enzyme processes $R$ and another set of kinetics when it processes $W$. The products and reactants end of the network correspond to the boundary between

the bulk phase and the vacuum. The kinetics of $R$ need to be such that the winding number is $w = 1$, localizing it at the products end, while the kinetics of $W$ need to have winding number $w = -1$, localizing it at the reactants end. Unlike the case of adaptation, the localization here is between one bulk phase and the vacuum and not between two bulk phases. Finally, adding multiple bulk phases in proofreading networks can allow for multiple discriminatory regimes in one network, including 'anti-proofreading' regimes[26].

## Discussion

Our results demonstrate that non-equilibrium systems can support topologically protected localized modes that resemble boundary modes found in topological insulators[10]. These protected modes can provide a general and compact framework to understand the robust functioning of microscopic non-equilibrium systems. Specifically, they elucidate how non-equilibrium fluxes can be used to create robust steady states with densities localized in preferred regions of phase space. While we have illustrated our results using idealized Markov state representation of certain biochemical processes, we anticipate that our results will find broad applicability in other areas such as dissipative self assembly and self organization in active matter systems. These are promising directions for future research.

## Methods

**Biophysical model for chemotaxis adaptation.** We consider a commonly used idealized biophysical model[20] for chemotaxis adaptation in *E. coli*. The dynamics of chemotaxis adaptation in *E. coli* can be described by specifying the activity $a$, and methylation level $m$ of the concentration sensing complex of proteins[15]. In the most typical modelling approach, the rates for transitions between states of different activity $a$ are derived from a chemoattractant concentration dependent-free energy landscape[21,22,27],

$$f(a, m) = -(a - 1/2)[mE - SM] \qquad (9)$$

where $m$ denotes the methylation level, the parameter $S$ is related to the logarithm of the chemoattractant concentration, $E$ set the energy scale of interaction between the activity $a$ and methylation level $m$, and $M$ is the total number of methylation levels. The rates of transition between active and inactive states, denoted by $\nu_0$ and $\nu_1$, respectively, couple to the concentration of the chemoattractant according to

the free energy functional above, In accordance with most biophysical models, we assume that the rates of transition between active and inactive states is given by

$$\nu_0 = \frac{\tau_a^{-1}}{1 + \exp \Delta_a f}, \quad \nu_1 = \frac{\tau_a^{-1}}{1 + \exp(-\Delta_a f)} \tag{10}$$

where $\tau_a$ is a constant that sets the timescales for transitions along the activity axis and $\Delta_a f = f(0, m) - f(1, m)$ is the free energy difference between states of different activity.

Adaptation is an active non-equilibrium process[5]; accordingly, methylation and de-methylation rates along the horizontal axis are determined by a non-equilibrium driving force $G$ in addition to the free energy landscape;

$$\ln \frac{r_a(m)}{l_a(m)} = f(a, m) - f(a, m+1) - (a - 1/2)G \tag{11}$$

The rates of transition along the methylation axis are set by a convention similar to that in equation (10) with a timescale $\tau_m$. In this class of biophysical modes, the rates of transition along the methylation axis are independent of methylation level and ligand concentration[15].

The transition rates along the activity axis specified by equation (10) and free energy surfaces like equation (9) generically are sigmoid functions of the methylation level $m$. The parameter $S$ specified by the ligand concentration sets the location of the crossover region. This generic sigmoidal profile for the rates of transitions along the activity axis, and the methylation level independent kinetics along the methylation axis establishes the similarity between the chemotaxis adaptation network and the network we constructed in Fig. 1a. Specifically, the two vertical levels denote the active and inactive states of the chemotaxis network and the horizontal levels denote the methylation level of the complex. The minimal biophysical model for adaptation can hence be viewed as a combination of two *bulk* networks with an interface between them. The location of the interface is set by the ligand concentration through the ligand concentration dependent parameter $S$.

**Linear algebra calculations and numerical simulations.** The numerical eigenvalue calculations were performed on Mathematica using the ARNOLDI method for sparse matrices. The numerical simulations were performed on Mathematica using standard kinetic MonteCarlo algorithms.

**Data availability.** The data used in this paper will be hosted on https://github.com/svaikunt/topological-modes.

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

## Acknowledgements

We gratefully acknowledge conversations with Aaron Dinner, Todd Gingrich, Mike Rust, Pankaj Mehta and Tom Witten. A.M. and S.V. were funded by NSF DMR-MRSEC 1420709. S.V. was also funded by a grant from the Army Research Office under grant number W911NF-16-1-0415.

## Author contributions

S.V. and A.M. designed the work, contributed analytical and numerical tools, performed the calculations and wrote the paper.

## Additional information

**Competing financial interests:** The authors declare no competing financial interests.

**Publisher's note**: 

