## [Peer Review File · Nature Communications]

Reviewer #2 (Remarks to the Author):

This manuscript describes some theoretical results on "edge modes" in master equation system with boundaries. It has been known that these modes appear in topological insulators as well as in isostatic mechanical systems. These modes are confined near the boundaries, because all bulk modes are "gapped out". It is completely natural to expect that master equation systems, which is mathematically closely related to the former systems, also exhibit these edge modes. As pointed out, these modes may have interesting implications in various non-equilibrium processes.

The topic is hence interesting. The current version of manuscript is however not suitable for publishing. The derivation of main results is not sufficiently sound. For example, 1) the index theorem Eq. (4) is apparently different from the corresponding result in ref. [10]. It therefore needs an explicit proof, instead of simply invoking Ref.[10]. 2) It is not clear why vanishing current along x axis means that the probability is localized (page 2). 3) There are also many typos and errors in writing. I feel that the manuscript needs a major rewriting before it can be considered for publication.

Reviewer #3 (Remarks to the Author):

The authors demonstrate the existence of topologically protected modes (akin to those discovered in topological insulators) in Markov state models of broad relevance to out of equilibrium soft matter systems and biological networks. I find this study to be very original and deserving of publication in Nature Communications because (to the best of my knowledge) no prior publications exist that explore topological modes in this context. Realizations of these ideas do exist outside quantum physics, most notably in optics and mechanics (cited by the authors) but there are significant differences that the authors elucidate in the main text and even more so in the SI. I also find the methodology of this study convincing and thorough because it blends theoretical analysis and simulations. The conclusions are sound and novel and the paper is lucidly written.

I do have a few minor points to raise:

1) In my opinion in the first section it could be useful to introduce the Markov model for a single bulk system first before discussing the interface between two. Right now the interface is already mentioned on line four a bit abruptly.

2) I find the discussion of the physical interpretation of the zero modes of $W^T(\lambda)$ too concise. It is an interesting point. If I understand correctly they should correspond to the so called states of self stress in the topological mechanical systems. Is that right? What role do they play in this context?

3) The authors emphasize how the agreement between numerics and theory is remarkable given the fact that the simulations have disorder in it. In a sense, this robustness is expected and a further validation of their assertion that these systems are topologically protected. However, usually (meaning in electronic systems) if the disorder is increased beyond a certain threshold the protection is lost, loosely speaking the system is no longer gapped. The effect of disorder depends also on whether it is compatible with the symmetry class of the system. Is there something similar at work here?

4) the following statement is very interesting but can the authors be more specific about the time scales required?

"The band gap in the topologically nontrivial regime imposes a constraint on the time scales required to perturb the topologically protected mode and provides another basis for robustness."

Perhaps this point is related to question 3) above?

4) I was confused by this statement at the very end of the paper

"Unlike the case of adaptation, the localization here is between one bulk phase and the vacuum and not between two bulk phases."

Can the localized modes also exist between two bulk phases as long as they are characterized by different topological invariants?

REVIEWERS' COMMENTS:

Reviewer #2 (Remarks to the Author):

The authors have addressed all my concerns so I think the revised manuscript is publishable in NC.

Reviewer #3 (Remarks to the Author):

The authors have addressed all my questions and concerns. I recommend publication of this manuscript.

Reviewer #2 (Remarks to the Author):

This manuscript describes some theoretical results on "edge modes" in master equation system with boundaries. It has been known that these modes appear in topological insulators as well as in isostatic mechanical systems. These modes are confined near the boundaries, because all bulk modes are "gapped out". It is completely natural to expect that master equation systems, which is mathematically closely related to the former systems, also exhibit these edge modes. As pointed out, these modes may have interesting implications in various non-equilibrium processes.

The topic is hence interesting. The current version of manuscript is however not suitable for publishing. The derivation of main results is not sufficiently sound. For example, 1) the index theorem Eq. (4) is apparently different from the corresponding result in ref. [10]. It therefore needs an explicit proof, instead of simply invoking Ref.[10]. 2) It is not clear why vanishing current along x axis means that the probability is localized (page 2). 3) There are also many typos and errors in writing.

I feel that the manuscript needs a major rewriting before it can be considered for publication.

We thank the referee for these remarks. We have rewritten the manuscript to address the referee's concerns.

1) We have pointed out that the first section of the SI contains an explicit proof for a network with a geometry similar to that in Fig 1 a. The only minor difference in this explicit proof is that the network has an interface with vacuum as opposed to another network. The next section of the SI has a proof based on Ref 10. We have now also clarified in the SI how the proof used in Ref 10 can be adapted to our system. Specifically, our constructed Hamiltonian anti commutes with a block diagonal version of the Pauli z matrix. We can hence write the index theorem in a form very similar to that in Ref 10. We have also updated the graphs in the SI and main text to provide more conclusive numerical evidence for Eq 4 (and the other results).

2) We have clarified the connection between zero current and localization of probability. The subsection D of section 1 in the SI has a proof for this statement. Briefly, topologically non-trivial far from equilibrium bulk networks support a non-zero current along the horizontal axis in their steady state. Requiring that this current is zero in the composited network in Fig 1c composed of two bulk regions results in an exponentially localized probability.

3) We have thoroughly reviewed the manuscript to remove typos and errors in writing. We have also rewritten a portion of the manuscript in response to Referee 2 and Referee 3's comments. We hope that this has improved the readability of the manuscript.

Reviewer #3 (Remarks to the Author):

The authors demonstrate the existence of topologically protected modes (akin to those discovered in topological insulators) in Markov state models of broad relevance to out of equilibrium soft matter systems and biological networks. I find this study to be very original and deserving of publication in Nature Communications because (to the best of my knowledge) no prior publications exist that explore topological modes in this context. Realizations of these ideas do exist outside quantum physics, most notably in optics and mechanics (cited by the authors) but there are significant differences that the authors elucidate in the main text and even more so in the SI. I also find the methodology of this study convincing and thorough because it blends theoretical analysis and simulations. The conclusions are sound and novel and the paper is lucidly written.

I do have a few minor points to raise:

1) In my opinion in the first section it could be useful to introduce the Markov model for a single bulk system first before discussing the interface between two. Right now the interface is already mentioned on line four a bit abruptly.

We thank the referee for pointing this out. In the revised manuscript, we have included text to better motivate the Markov state model.

2) I find the discussion of the physical interpretation of the zero modes of $W^T(\lambda)$ too concise. It is an interesting point. If I understand correctly they should correspond to the so called states of self stress in the topological mechanical systems. Is that right? What role do they play in this context?

It has been established in other contexts that the eigenmodes of W^T have dynamical information related to first passage times etc. In this sense, they indeed carry information complimentary to the information carried by the eigenvector of W , namely the steady state probability. We have included a short discussion in the text on this issue.

3) The authors emphasize how the agreement between numerics and theory is remarkable given the fact that the simulations have disorder in it. In a sense, this robustness is expected and a further validation of their assertion that these systems are topologically protected. However, usually (meaning in electronic systems) if the disorder is increased beyond a certain threshold the

protection is lost, loosely speaking the system is no longer gapped. The effect of disorder depends also on whether it is compatible with the symmetry class of the system. Is there something similar at work here?

Yes, we anticipate that there is something similar at work here. We are fundamentally concerned with Markov state jump processes. If the kinetic rates for hops across a particular set of nodes in the bulk is dramatically altered due to disorder, we can expect transient spatial localization of the probability in such bulk regions (see for instance Ref Vaikuntanathan 2014). Such features might indeed lead to a loss of protection in the bulk for a finite network.

4) the following statement is very interesting but can the authors be more specific about the time scales required?

"The band gap in the topologically nontrivial regime imposes a constraint on the time scales required to perturb the topologically protected mode and provides another basis for robustness."

Perhaps this point is related to question 3) above?

We thank the referee for bringing this point up. Since the non zero eigenvalues of the master equation rate matrix set relaxation timescales, the band gap in the topologically nontrivial regime implies that perturbations away from the localized steady state are suppressed in a finite time τ related inversely to the band gap. This feature provides another basis for robustness. We have clarified this connection in the main text.

Apart from this, we anticipate that crossing the band gap is akin to crossing a dynamical phase transition. This connection is made by regarding the cumulant generating function for currents as a generalized dynamical free energy. We expect a kink or a discontinuity in the slope of the cumulant generating function, at points λ_{pm} to signal a first order dynamical phase transition (see Ref 19 of main text) . In this context, crossing the transition will be accompanied by hysteresis and other finite size effects. The system has to be quenched using an infinitesimally slow process to take it from the protected to the unprotected state through the transition. These statements clearly need more work/data for verification. Hence, we have not provided this argument in the main text. Below, we include some preliminary results in which we investigated the statistics of entropy production for the ladder network in its topologically protected state. We indeed see the presence of a fat tail in the distribution of entropy production consistent with the presence of a first order phase transition

Here σ denotes entropy production rate and $I(\sigma)$ denotes the large deviation functional for entropy production. The rate function $I(\sigma)$ is proportional to $-\ln(P(\sigma))$ where $P(\sigma)$ is the probability of observing a certain entropy production rate.

4) I was confused by this statement at the very end of the paper

"Unlike the case of adaptation, the localization here is between one bulk phase and the vacuum and not between two bulk phases."

Can the localized modes also exist between two bulk phases as long as they are characterized by different topological invariants?

Yes, localized modes can exist as long as the topological invariants are different.